# Hierarchical-Latent Generative Models are Robust View Generators for Contrastive Representation Learning

## Abstract

A growing area of research is exploiting pre-trained generative models as a data source for contrastive representation learning, generating anchors and the associated positive views through perturbations of the latent codes. In this study, we make significant advances in this field by formalizing the properties of a specific category of generative models, which we term Hierarchical-Latent. We show how the intrinsic properties of these models can successfully be used to create robust views for contrastive learning, outperforming not only previous methods' performance but also surpassing classic approaches trained with genuine real data. The proposed framework is evaluated on different generators and contrastive learning techniques, also investigating the effects of employing a discriminator to filter out low-quality images. Eventually, we test continuous sampling in our experiments, where the generator dynamically samples new synthetic data during contrastive training of the encoder, showing competitive or faster training time with respect to a real-data approach, while allowing a virtually unlimited training set.

## 1 Introduction

Self-supervised learning techniques (SSL) allow the extraction of meaningful information from vast amounts of unlabeled data, drastically reducing or even eliminating the need for costly manual annotations. Depending on the purpose of the trained networks, these approaches can be broadly classified into two main categories: generative and contrastive (Liu et al., 2021).

The former methods include various approaches, like Variational Autoencoders (VAE) Kingma & Welling (2014), Generative Adversarial Networks (GAN) Goodfellow et al. (2014), or Normalizing Flows Rezende & Mohamed (2015) (NF), and have received growing interest due to their ability in approximating real data distributions (Rombach et al., 2022; Yu et al., 2022). Under this perspective, generators can be seen as compact representations of such distributions, and can thus be used as a data source for training discriminative networks (Besnier et al., 2020; Sariyildiz et al., 2023).

On the other hand, contrastive techniques like Chen et al. (2020b); Caron et al. (2020) aim to learn an embedding function $f$ mapping similar images $\mathbf{x}_1, \mathbf{x}_2$ (referred to as positives) to nearby latent representations $\mathbf{e}_1, \mathbf{e}_2$, ensuring that dissimilar samples (negatives) are pushed apart in the latent space. While approaches like Grill et al. (2020); He et al. (2020) obtained good results even without the use of negatives, designing strong positives remains a significant challenge (Figure 1, Left).

In this paper, we first give a formal definition for a specific category of image generation models, termed Hierarchical-Latent (HL), and observe how the multiple latent spaces of these networks influence the final image at different levels, exhibiting a general global-to-local pattern. Then, inspired by Jahanian et al. (2021); Li et al. (2022b), we unify the two self-supervised categories discussed above by employing these generative models both as a data source and to obtain positive views for contrastive representation learning (Figure 1, Right).

Specifically, the generative process of HL models is influenced by multiple random latent vectors, that enter the network progressively, at different blocks (hierarchy levels). Moreover, global aspects of the image are addressed in early blocks and finer details are only subsequently refined, implying that there exists a direct relationship between a network block and the perturbation that can be

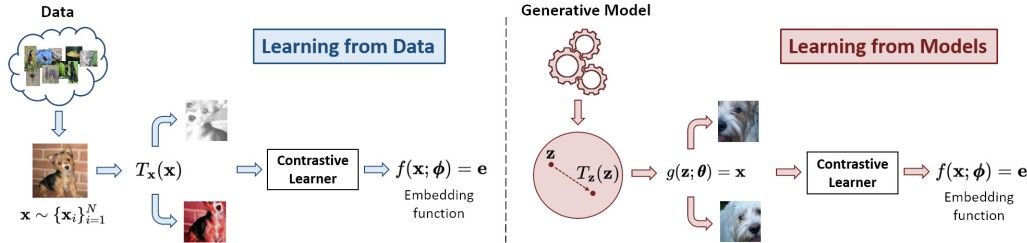

Figure 1: Contrastive learning aims at optimizing an embedding function $f(\mathbf{x}; \boldsymbol{\phi}) = \mathbf{e}$ with parameters $\boldsymbol{\phi}$. **Left**: in the "Learning from Data" paradigm, positive views are usually obtained through manually designed pixel-space augmentations $T_{\mathbf{x}}$, applied on a finite set of images $\mathbf{x} = \{\mathbf{x}_i\}_{i=1}^N$ approximating the real data distribution. **Right**: in the "Learning from Models" paradigm, a generator with parameters $\boldsymbol{\theta}$: $g(\mathbf{z}; \boldsymbol{\theta}) = \mathbf{x}$ provides a compact approximation of the real data distribution. Positive views can be obtained with small perturbations of the anchors $T_{\mathbf{z}}(\mathbf{z})$ in the generator's latent space.

applied to its corresponding latent vector without affecting the final image semantics. In Section 3 we extensively discuss this phenomenon: we propose to learn (for each latent space - indexed by $p$), an appropriate non-linear function $T_{\mathbf{z}}^p$ providing the maximum distance $\delta^p$ that can be taken from a point $\mathbf{z}^p$ while maintaining semantic consistency of the generated image. These so-called *walkers* are used for a Monte Carlo estimation on the $\delta$ values, which are shown to increase for higher hierarchies. In the context of positive view generation, we take advantage of these findings and apply different perturbations to each hierarchical latent vector to change only the semantically non-relevant characteristics of the initial anchors.

In the experimental part, we train contrastive encoders utilizing well-known frameworks (SimCLR Chen et al. (2020b) and SimSiam Chen & He (2021)) and consider two HL model scenarios: a BigBiGan (Donahue & Simonyan, 2019) pre-trained on the general and diverse ImageNet-1K (Deng et al., 2009) dataset, and a StyleGan2 (Karras et al., 2020) pre-trained on the fine-grained LSUN Cars (Yu et al., 2015). The encoders' generalization capabilities are evaluated through linear classification on multiple downstream datasets and on Pascal VOC Everingham et al. (2010) for the detection task. *Our results surpass both state-of-the-art methods and training with real data.*

Additionally, to further enhance diversity in the generations and reduce the accuracy gap that typically appears between classifiers trained with synthetic vs. real data Ravuri & Vinyals (2019), we propose continuous sampling to increase the total number of training images. Differently from previous methods (Besnier et al., 2020; Sariyildiz et al., 2023; Lampis et al., 2023), that sample additional data "offline", we propose an "online" approach to generate a continuous stream of new images directly during training. We also show that a fast sampling generator (e.g. a GAN) achieves competitive or faster training time duration when compared to standard data loading techniques.

To sum up, our contributions are the following: 1) We give a definition for the Hierarchical-Latent (HL) category of image generation models, formalizing their properties. 2) We show how such models can be applied to generate anchors and positives for contrastive representation learning, surpassing state-of-the-art methods and the real training data case. 3) We further show how employing continuous sampling can increase the total number of seen images while maintaining a comparable or faster overall training time with respect to standard data loading techniques.

## 2 BACKGROUND AND PROBLEM FORMULATION

**Self-Supervised representation learning (SSRL).**    The focus of the proposed methodology is on positive view sampling, and it can thus be integrated into any SSRL framework. In the experiments, we validate the efficacy of our approach through two different pipelines, SimCLR Chen et al. (2020b) and SimSiam Chen & He (2021), as they are widely adopted techniques in the literature.

The former method obtains positives through a diverse range of manually designed pixel-space augmentation $T_{\mathbf{x}}$, which can be applied (after image sampling) on top of any generators' latent space perturbation $T_{\mathbf{z}}$, as in Jahanian et al. (2021); Li et al. (2022b). The final anchor and positive pair $\mathbf{x}, \mathbf{x}^+$ are used to minimize the following InfoNCE loss Oord et al. (2018) and thus learn the embedding function $f$ (parameters $\boldsymbol{\phi}$ omitted in the formula):

$$\mathcal{L}_{\text{NCE}} = \mathbb{E}_{\mathbf{x}, \mathbf{x}^+} \left[ -\log \left( \frac{e^{\text{sim}(f(\mathbf{x}), f(\mathbf{x}^+))/\tau}}{e^{\text{sim}(f(\mathbf{x}), f(\mathbf{x}^+))/\tau} + \sum_{k=1}^{K} e^{\text{sim}(f(\mathbf{x}), f(\mathbf{x}^k))/\tau}} \right) \right] \quad (1)$$

where $\text{sim}$ denotes the cosine similarity operator, $\tau$ is a temperature parameter and $K$ the number of negative samples in a minibatch.

The latter approach, SimSiam, introduces a stop-gradient operator to prevent the degeneration to trivial solutions, where all images share the same representation. This allows SimSiam to avoid the use of negative samples, making it more suitable to investigate the effects of positive view generation methods. More in detail, two views $\mathbf{x_1}, \mathbf{x_2}$ are processed by a learnable encoder $f$ and an additional Multi-Layer Perceptron $h$ (with parameters $\boldsymbol{\phi}$ and $\boldsymbol{\theta}$, respectively). The obtained representations are defined as $\mathbf{z_i} := f(\mathbf{x_i}, \boldsymbol{\phi})$ and $\mathbf{p_i} := h(\mathbf{z_i}, \boldsymbol{\theta})$, while the final loss function is given by the symmetric comparison:

$$\mathcal{L} = \frac{1}{2} \text{sim}(\mathbf{p_1}, \text{stopgrad}(\mathbf{z_2})) + \frac{1}{2} \text{sim}(\text{stopgrad}(\mathbf{z_1}), \mathbf{p_2}) \quad (2)$$

where $\text{sim}$ is the negative cosine similarity and $\text{stopgrad}$ the stop-gradient operator.

**Robust views for contrastive learning.** When defining a set of augmentations for contrastive learning (both manually in the pixel space or through a generative model) it is fundamental to consider which transformations the trained encoder should be invariant to (Xiao et al., 2020). As outlined in the following "Infomin" principle, this means that optimal views can be achieved only if the downstream task is known in advance, since each transformation is suited only for some specific problems.

**Proposition 2.1.** *(Optimal Views for Contrastive Learning - Tian et al. (2020))*
*Given a downstream task $\mathcal{T}$ with label $\mathbf{y} \in \mathcal{Y}$, the optimal views created from data $\mathbf{x}$ are*

$$(\mathbf{v_1^*}; \mathbf{v_2^*}) = \underset{\mathbf{v_1}; \mathbf{v_2}}{\arg\min} \, I(\mathbf{v_1}; \mathbf{v_2}) \quad \text{subject to} \quad I(\mathbf{v_1}; \mathbf{y}) = I(\mathbf{v_2}; \mathbf{y}) = I(\mathbf{x}; \mathbf{y})$$

*In other terms, the mutual information between optimal views is minimized to contain only the task-relevant information $I(\mathbf{v_1^*}; \mathbf{v_2^*}) = I(\mathbf{x}; \mathbf{y})$, removing all nuisance information, $I(\mathbf{v_1^*}; \mathbf{v_2^*}|\mathbf{y}) = 0$.*

The aforementioned concept of optimal views is extended in Li et al. (2022b), specifically in terms of distances between generators' latent vectors. In the following, we give a reformulation that is better suited for the understanding of our work.

**Proposition 2.2.** *(Semantic Consistency in the Latent Space - Li et al. (2022b) (reformulation))*
*Let $g(\mathbf{z}, \boldsymbol{\theta}) = \mathbf{x}$ be a generative model with parameters $\boldsymbol{\theta}$ that maps from latents $\mathbf{z} \in \mathbb{R}^n$ to images $\mathbf{x} \in \mathbb{R}^d$. Consider a downstream task $\mathcal{T}$ with label $\mathbf{y} \in \mathcal{Y}$, and some distance metric $\text{dist}$ defined in the generator's latent space. Then, for any random latent vector $\mathbf{z}_i$, the term $\delta_i$ indicates the maximum distance in the latent space of $g$ to ensure semantic consistency for task $\mathcal{T}$:*

$$\delta_i = \max_{\mathbf{z}'} \text{dist}(\mathbf{z}_i, \mathbf{z}') \text{ s.t. } F_{\mathcal{T}}(g(\mathbf{z}_i, \boldsymbol{\theta})) = F_{\mathcal{T}}(g(\mathbf{z}', \boldsymbol{\theta})) = \mathbf{y}$$

*where $\mathbf{z}' \in \mathbb{R}^n$ is a generic vector in the generator's latent space and $F_{\mathcal{T}}$ is an oracle classification function for task $\mathcal{T}$, which assigns the true semantic label $\mathbf{y}$ to any image.*

Note that the term $\delta_i$ is associated with a corresponding latent $\mathbf{z}_i \in \mathbb{R}^n$ and thus an infinite number of $\delta$ terms exist for a specific generator's latent space and task $\mathcal{T}$. In the rest of this paper, we define as $\Delta_{\mathcal{T}}$ the distribution over all $\delta$ terms for task $\mathcal{T}$.

From Propositions 2.1 and 2.2 follow that the generated optimal views $g(\mathbf{z}_1, \boldsymbol{\theta}), g(\mathbf{z}_2, \boldsymbol{\theta})$ for task $\mathcal{T}$ can be derived from $\mathbf{z}_1, \mathbf{z}_2$ only if $\text{dist}(\mathbf{z}_1, \mathbf{z}_2) = \delta_1$. In our study, *we assume $\mathcal{T}$ to be unknown in advance*. As a consequence, differently from Li et al. (2022b), we refer to our views as **robust** rather than optimal, meaning that the trained encoders achieve good results on different downstream tasks, rather than optimal scores on a single problem.

**View generation with latent perturbations.** In the existing literature, two prominent approaches address the task of defining perturbations in a pre-trained generator's latent space to obtain useful views for contrastive learning. In this study, we consider them as baselines in the experimental phase, while extending their definition for HL generators.

More in detail, Jahanian et al. (2021) were the first to address this challenge and explored various solutions. Among these, the most favorable result was obtained by simply adding Gaussian noise to the sampled anchor. Mathematically, considering an anchor $\mathbf{z}$ in the generator's latent space, the transformation responsible for generating the positive samples is expressed as $T_{\mathbf{z}}(\mathbf{z}) = \mathbf{z} + \mathbf{w}_{\text{rand}}$, where $\mathbf{w}_{\text{rand}} \sim \mathcal{N}^t(\mu, \sigma, t)$ denotes a truncated Gaussian distribution with truncation $t$. Throughout the rest of this paper, we refer to this technique as **random** sampling of the positives.

On the other hand, Li et al. (2022b) introduced an adversarial strategy to learn the optimal perturbation $T_{\mathbf{z}}$ for each instance. Semantic consistency is ensured by the minimization of InfoNCE loss between the generated views, which corresponds to maximizing a lower bound on mutual information (Oord et al., 2018). The final positive sample is still computed as the sum with respect to the anchor: $T_{\mathbf{z}}(\mathbf{z}) = \mathbf{z} + \mathbf{w}_{\text{learn}}$, but $\mathbf{w}_{\text{learn}}$ is a non-linear learnable function (*walker*) optimized through a small contrastive encoder $f^*(\mathbf{x}, \boldsymbol{\phi})$ (parameters $\boldsymbol{\phi}$ omitted in the formula):

$$\max_{T_{\mathbf{z}}} \min_{f^*} \mathcal{L}_{\text{NCE}}\big(f^*(g(\mathbf{z})), f^*(g(T_{\mathbf{z}}(\mathbf{z})))\big), \text{ s.t. } \text{dist}(\mathbf{z}, T_{\mathbf{z}}(\mathbf{z})) \leq \delta \quad (3)$$

In other terms, the training aims at learning the perturbation $T_{\mathbf{z}}(\mathbf{z})$ for each $\mathbf{z}$ such that $\text{dist}(\mathbf{z}, T_{\mathbf{z}}(\mathbf{z})) \approx \delta$, where $\delta$ is defined for each instance $i$ as in Proposition 2.2. After this process, the encoder $f^*$ is discarded, and the learned $T_{\mathbf{z}}$ can be used to generate views from the sampled anchors. In the rest of this paper, we refer to this method as the **learned** sampling of positives.

## 3 POSITIVE SAMPLING VIA HIERARCHICAL-LATENT GENERATIVE MODELS

### 3.1 DEFINITIONS AND ASSUMPTIONS

In this Section, we provide a formal definition for a specific category of generative models, termed Hierarchical-Latent (HL). This family extends any generator $g(\mathbf{z}, \boldsymbol{\theta}) = \mathbf{x}$ mapping latent variables $\mathbf{z} \in \mathbb{R}^n$ to images $\mathbf{x} \in \mathbb{R}^d$, including models like Generative Adversarial Networks (GANs), Variational Autoencoders (VAEs) or Normalizing Flows (NFs).

**Definition 3.1.** *(Hierarchical-Latent Generative Models)*
*A Hierarchical-Latent (HL) generative model $g(\mathbf{z}^0, \mathbf{z}^1, \ldots, \mathbf{z}^{n-1}, \boldsymbol{\theta}) = \mathbf{x}$ is a deep neural network with parameters $\boldsymbol{\theta}$ that samples new data $\mathbf{x}$ by incorporating multiple random latent variables $\{\mathbf{z}^0, \mathbf{z}^1, \ldots, \mathbf{z}^{n-1}\}$ at different progressive blocks:*

$$g : \mathbb{R}_0^{m_0} \times \mathbb{R}_1^{m_1} \times \cdots \times \mathbb{R}_{n-1}^{m_{n-1}} \to \mathbb{R}^d$$
$$g := l_{[n-1]}(\mathbf{z}^{n-1}, l_{[n-2]}(\mathbf{z}^{n-2}, \ldots l_{[0]}(\mathbf{z}^0) \ldots))$$

*where $l_{[i]}$ is the $i^{th}$ block of the generator and $\mathbf{z}^i$ its corresponding latent vector.*

In the rest of this study, we assume that latent hierarchies share the same dimensionality (meaning that $m_0 = m_1 = \cdots = m$), as this is the case in most practical applications.

Hierarchical latent variables play a crucial role in capturing diverse abstraction levels and controlling the generative process. Specifically, given the initial input $\mathbf{z}^0$, it influences the output of all the subsequent layers, and thereby is supposed to have a significant impact on the overall structure and global properties of the generated image. On the other hand, $\mathbf{z}^{n-1}$ only enters at the last block, and thus it typically models only some fine-grained details. In the following, we formulate such a general intuition in terms of semantic consistency in the latent space (as defined in Proposition 2.2):

**Assumption 3.1.** *(Semantic Consistency relation in HL Generative Models)*
*Consider $\mathcal{T}$ as the general task of finding **robust** views that preserve the relevant semantic content of an image while altering all the remaining information. Let $g : \mathbb{R}_0^m \times \mathbb{R}_1^m \times \cdots \times \mathbb{R}_{n-1}^m \to \mathbb{R}^d$ be an HL generative model with $n$ input variables. Then, for any generic instance $\{\mathbf{z}_i^0, \mathbf{z}_i^1, \ldots, \mathbf{z}_i^{n-1}\}$ and its associated semantic consistency terms $\{\delta_i^0, \delta_i^1, \ldots, \delta_i^{n-1}\}$ for task $\mathcal{T}$ (see Proposition 2.2), the following relation is assumed to hold:*

$$\delta_i^p < \delta_i^q \quad \forall p, q \text{ such that } p < q$$

In other terms, Assumption 3.1 states that the perturbation that can be applied to the latent variable $\mathbf{z}_i^p$ without altering the semantic content of the generated image is smaller than the one applicable to any

$\mathbf{z}_i^q$, when $p < q$ and for any instance $i$. Nevertheless, it should be noted that the "*relevant semantic content*" of an image always depends on the final downstream task (see Proposition 2.1), which we consider to be unknown. Hence, given the general definition for $\mathcal{T}$, the assumption's validity and its applications to view generation are further discussed in Section 3.2 for each scenario considered within this study.

## 3.2 VIEW GENERATION

For the purpose of robust view generation, we study two different scenarios. The former considers contrastive encoders trained on a large and diverse number of classes, specifically ImageNet-1K Deng et al. (2009) for real data and BigBiGan Donahue & Simonyan (2019) as an HL generator (trained on the same dataset). The latter is a fine-grained case, employing a StyleGan2 Karras et al. (2020) generator trained on LSUN Cars Yu et al. (2015). Note that both these models have also been employed in recent works (Jahanian et al., 2021; Li et al., 2022b), but without exploring the HL structure and instead treating the latent vectors as a whole, i.e., non-hierarchically. In contrast, our work applies both *random* and *learned* perturbations separately at different latent levels, enabling a better utilization of the distinct hierarchies of HL models. We observe how (depending on the scenario/generator) each of these transformations can be tuned to act on one or more aspects of the output sample, influencing only the desired characteristics.

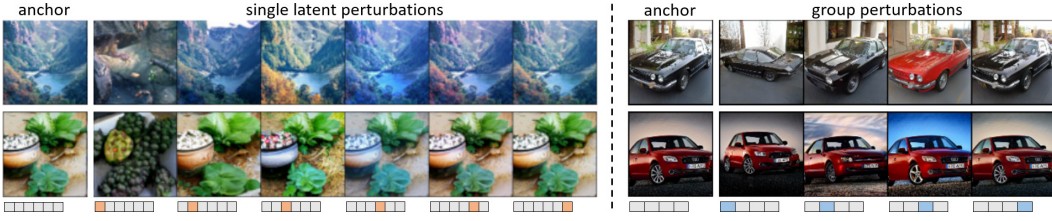

Figure 2: Examples of views generated by adding the **same** noise vector $\mathbf{w}$ to different hierarchical levels. **Left**: two anchor images and possible views generated by the perturbations of BigBiGan's 6 latent hierarchies $\{\mathbf{z}^0, \mathbf{z}^1, \ldots, \mathbf{z}^5\}$, represented as the 6 elements' vector at the bottom. The orange element indicates the applied perturbation $T_{\mathbf{z}}^p(\mathbf{z}^p) = \mathbf{z}^p + \mathbf{w}$ for each level $p$. **Right**: generated anchors and views by StyleGan2, which has 16 hierarchal levels, grouped into 4 sets and represented as the 4 elements' vector at the bottom. The blue element indicates the perturbed group.

The BigBiGan model possesses 6 latent hierarchies, each of length 20: $\{\mathbf{z}^0, \mathbf{z}^1, \ldots, \mathbf{z}^5\}$. Figure 2 (Left) shows how the same noise vector $\mathbf{w}$ influences the final image, if separately summed to each $\mathbf{z}^p$. In particular, it clearly appears that the first latent $\mathbf{z}^0$ mainly models the semantic aspects, while subsequent hierarchies modify more subtle details, like small shaping and colors. To better determine how each of these "chunks"[1] impacts the final generation, we propose to estimate the corresponding semantic consistency distributions $\{\Delta_{\mathcal{T}}^0, \Delta_{\mathcal{T}}^1, \ldots, \Delta_{\mathcal{T}}^5\}$ employing a Monte Carlo simulation. Specifically, we first train 6 different non-linear *walkers* $T_{\mathbf{z}}^p$ optimized for Equation (3), each acting on a single $\mathbf{z}^p$ and aiming to identify for each instance $i$ and hierarchy $p$ the maximum value of $\delta_i^p = \text{dist}(\mathbf{z}_i^p, T_{\mathbf{z}}^p(\mathbf{z}_i^p))$ that still allows minimizing the InfoNCE loss (Equation 1) on the encodings derived from the generated output views. After training, we perform Monte Carlo sampling of 1M anchors and use the pre-trained *walkers* to compute each $\delta_i^p = \text{dist}(\mathbf{z}_i^p, T_{\mathbf{z}}^p(\mathbf{z}_i^p))$, where dist is the Euclidean distance.

Table 1 (Left) reports the computed mean and standard deviation values of each estimated $\Delta_{\mathcal{T}}^p$, as well as the final InfoNCE loss value and total number of training samples for the corresponding *walker* $T_{\mathbf{z}}^p$. We notice that the average perturbation (estimated mean) needed to achieve the same semantic shift (InfoNCE loss value) in each $\mathbf{z}^p$ increases across hierarchies, supporting Assumption 3.1. This trend degenerates for $\mathbf{z}^5$, where even with extensive training (2260K samples), the *walker* fails to learn meaningful perturbations, maintaining very low loss values. In this latter case, visualizations show that some perturbed images remain identical to the anchor while others are completely black.

Moreover, the low mean value estimated for chunk 0 confirms that this hierarchy predominantly models the semantics of generated images, setting a strong upper bound when applying perturba-

---

[1]In the following, we informally refer to each latent hierarchy $\mathbf{z}^p$ also as a "chunk".

Table 1: **Left**: estimated mean ($\mu$) and standard deviation ($\sigma$) of $\Delta_{\mathcal{T}}^p$ for each hierarchy $p$. In each row, we also report for the corresponding *walker* the final loss value and the number of seen training samples. **Right**: the same computations obtained through a Monte Carlo estimation using two different pre-trained *walkers*, one acting on all hierarchies at once, the other ignoring the first chunk.

| chunk | single chunk | | | | chunk | all chunks | | > 0 chunks | |
|---|---|---|---|---|---|---|---|---|---|
| | Loss (InfoNCE) | train samples (K) | mean $\mu$ | std $\sigma$ | | mean $\mu$ | std $\sigma$ | mean $\mu$ | std $\sigma$ |
| 0 | 1.09 | 30.0 | 0.67 | 0.21 | 0 | 0.43 | 0.07 | - | - |
| 1 | 1.04 | 96.0 | 3.63 | 1.18 | 1 | 0.27 | 0.04 | 1.85 | 0.42 |
| 2 | 1.05 | 180.0 | 6.97 | 1.85 | 2 | 0.60 | 0.09 | 2.44 | 0.56 |
| 3 | 1.02 | 214.0 | 13.00 | 7.08 | 3 | 0.36 | 0.05 | 1.52 | 0.34 |
| 4 | 1.05 | 276.0 | 21.22 | 13.68 | 4 | 0.36 | 0.05 | 1.05 | 0.23 |
| 5 | 0.14 | 2260.0 | 594.71 | 616.80 | 5 | 0.33 | 0.05 | 0.70 | 0.15 |

tions. In the experiments, we thus decide to leave this first latent unchanged and apply stronger transformations to the other hierarchies. Specifically, in the *random* case we increase the truncated Gaussian's standard deviation parameter (check Section 4 for precise values). In the *learned* scenario, we train two different *walkers*: the baseline acting on all chunks simultaneously (as proposed in Li et al. (2022b)) and one forced to preserve the first chunk. Table 1 (Right) reports the Monte Carlo estimation on the single hierarchies comparing these cases. Results confirm that keeping chunk 0 fixed enables the *walker* to learn stronger perturbations on the other chunks.

Regarding the fine-grained scenario, generating solely images of cars poses a huge challenge in defining what the "*relevant semantic content*" of an image is, since even minor shape variations can lead to a different downstream label. Moreover, StyleGan2 possesses a larger number of hierarchies, with latent vectors defined in a non-Gaussian space, known as $\mathcal{W}$ and derived from an appropriately trained Multi-Layer Perceptron network (Karras et al., 2019).

Our aim for this challenging fine-grained setup is to showcase how HL generators can still provide useful views and improve baseline augmentation methods. For this purpose, we divide StyleGan2's 16 chunks into 4 groups and observe how they influence the final output content (Figure 2 (Right)). We note in particular how the first group mainly impacts rotations, zoom, and minor shape aspects; the second one focuses on subject and background alterations, while the latter two mainly define colors. In the experiments, we tune both the *random* and *learned* methods to separately perturb each group, limiting the applied transformations for the first two sets of chunks and amplifying the others.

## 4 EXPERIMENTS

### 4.1 GENERAL SETTING AND PRELIMINARY STUDIES

For each generator, we train multiple ResNet-50 encoders using the SimSiam (Chen & He, 2021) framework. Additionally, for BigBiGan, we extend the code of Li et al. (2022b)[2] to test the proposed HL perturbations also using SimCLR (Chen et al., 2020b). The representation capabilities of the obtained HL encoders are compared against several methods: training on synthetic data without latent perturbations $T_{\mathbf{z}}$, the *random* and *learned* baselines not leveraging the HL structure, and the upper bound of using real data (1.28M images for ImageNet-1K Deng et al. (2009) and 893K images for LSUN Cars Yu et al. (2015), depending on the generator). Pixel space augmentations $T_{\mathbf{x}}$ are tested in various combinations: cropping and horizontal flipping, grayscale and color jittering, none of the previous or all of them (detailed ablation studies in Appendix C). In contrast to prior studies, we find that the $T_{\mathbf{z}}$ perturbations introduced by HL models can replace the color pixel space augmentations by producing more realistic modifications, and thus we do not apply these particular $T_{\mathbf{x}}$ transformations in the HL runs.

Following prior studies (Jahanian et al., 2021; Li et al., 2022b), the SimSiam experiments use $128 \times 128$ input image size, matching the output resolution of the BigBiGan generator. All encoders (regardless of the scenario) are trained for 100 epochs using SGD optimizer with momentum 0.9

---

[2]Code: https://github.com/LiYinqi/COP-Gen

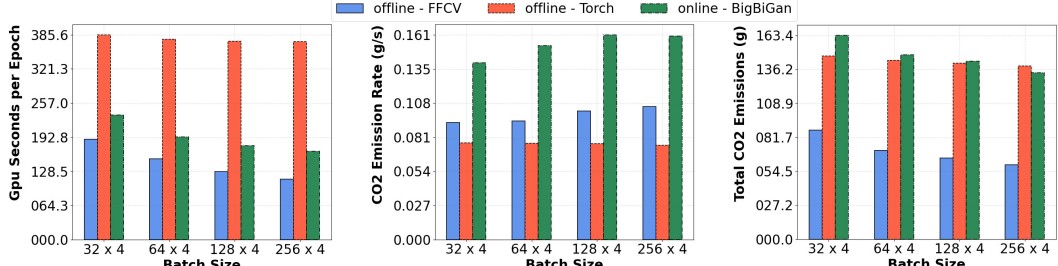

Figure 3: Total time (GPU seconds), $CO_2$ emissions rate (grams per second) and total $CO_2$ emissions (grams) for three different data loading procedures (offline `PyTorch`, offline `FFCV` and online with BigBiGan generator) and different batch sizes.

and weight decay $1 \times 10^{-4}$. The learning rate is set as $0.05 \times \text{BatchSize}/256$, with a cosine decay scheduler and an additional linear warmup for the first 10 epochs if BatchSize $\geq 1024$. Moreover, SimSiam experiments with synthetic data employ continuous sampling, where the generator outputs new batches directly on each GPU device. The number of training steps per epoch remains the same as in the real data case, but the total number of seen samples is greatly increased.

The use of continuous sampling is possible due to GANs' constant inference time. But how is overall training time affected? To answer this question, we trained a ResNet-18 model using SimCLR for 20 epochs on ImageNet-100 (a subset of ImageNet-1K), using 4 NVIDIA A100-SXM4-40GB GPUs and different batch sizes ($32 \times 4$, $64 \times 4$, $128 \times 4$, $256 \times 4$). The experiment has been run three times, specifically using the standard `PyTorch` Paszke et al. (2019) loader, the efficient `FFCV` Leclerc et al. (2023) loader (both with 8 workers), and the BigBiGan generator. Figure 3 displays our findings, reporting the mean GPU seconds per epoch, the $CO_2$ emissions rate, and the total $CO_2$ emissions estimated using the `CodeCarbon` library Schmidt et al. (2021) with default settings. Interestingly, continuous sampling proved significantly faster than the standard `torch` loader and only marginally slower than `FFCV`. In terms of $CO_2$ emissions rate, the use of BigBiGan led to higher energy consumption, due to intensive GPU usage. Nevertheless, in terms of total $CO_2$ the values remain comparable with the standard `torch` approach, suggesting that continuous sampling is not only feasible but may also be an interesting alternative to conventional techniques.

## 4.2 EXPERIMENTAL RESULTS

**BigBiGan and ImageNet-1K.** We first extend prior experiments with HL *random* and *learned* perturbations by following the SimCLR framework, code, data loading procedures, hyperparameters, and evaluation protocols outlined in Li et al. (2022b). Then, all results are computed also using SimSiam with a batch size of 1024. Anchors are sampled from $\mathcal{N}^t(0., 1., 2.)$, while the *random* runs use $\mathcal{N}^t(0., 0.05, 2.)$ in the baseline case and $\mathcal{N}^t(0., 1., 2.)$ in the HL scenario, keeping $\mathbf{z}^0$ fixed as indicated in Section 3.2. The standard deviation values are selected according to the best results over multiple attempts (detailed ablation studies in Appendix C). In the *learned* case, we maintain the same $T_{\mathbf{z}}$ used in Table 1 (Right), where the baseline and HL *walkers* have seen 60K and 120K samples, respectively. Additionally, we explore for HL runs how the quality of the generator affects overall performance, using a discriminator to screen out unrealistic samples from the training batches.

To assess the encoders' representation quality, we train linear classifiers on top of the pre-trained models and evaluate them on various classification datasets: ImageNet-1K, Birdsnap Berg et al. (2014), Caltech101 Fei-Fei et al. (2004), Cifar100 Krizhevsky et al. (2009), DTD Cimpoi et al. (2014), Flowers102 Nilsback & Zisserman (2008), Food101 Bossard et al. (2014), and Pets Parkhi et al. (2012). Classifiers are trained for 60 epochs with a batch size of 256, SGD optimizer, and a learning rate of 30.0 with cosine decay. We also evaluate on Pascal VOC Everingham et al. (2010) object detection using `detectron 2` Wu et al. (2019) to train a Faster-RCNN with the R50-C4 backbone. All layers are fine-tuned for 24000 iterations on `trainval07+12` split and evaluated on `test07`. Table 2 shows the results for ImageNet-1K and Pascal VOC evaluations, while Table 3 (Left) indicates the mean Top-1 accuracy for the SimSiam encoders, computed over the 7 transfer datasets (Appendix D reports the single runs, each of which has been replicated 5 times).

Table 2: Comparison of baselines and proposed HL perturbations on two contrastive frameworks (SimCLR and SimSiam). Metrics are Top-1 and Top-5 accuracy for linear classification on ImageNet-1K and average precision for detection on Pascal VOC. Bold indicates the best results for each group, underline the absolute best, and * indicates the baseline reported from Li et al. (2022b).

| Data | $T_{\mathbf{z}}$ | $T_{\mathbf{x}}$ | SimCLR | | | | | SimSiam | | | | |
|---|---|---|---|---|---|---|---|---|---|---|---|---|
| | | | ImageNet-1K | | Pascal VOC | | | ImageNet-1K | | Pascal VOC | | |
| | | | Top-1 | Top-5 | AP | $AP_{50}$ | $AP_{75}$ | Top-1 | Top-5 | AP | $AP_{50}$ | $AP_{75}$ |
| real | - | all | **49.4*** | **75.6*** | **52.9*** | **78.7*** | **58.5*** | 49.1 | **74.2** | 54.4 | **80.0** | **60.0** |
| synth | - | all | 41.6* | 66.6* | 51.0* | 77.2* | 55.8* | 32.2 | 56.5 | 51.6 | 78.2 | 57.0 |
| synth | random | all | 48.7* | 73.1* | 50.2* | 77.0* | 54.4* | 33.4 | 57.7 | 51.7 | 78.4 | 56.3 |
| synth | HL rand. | no col. | **53.7** | **77.2** | **53.3** | **79.5** | **58.5** | 42.5 | 67.7 | **54.3** | **79.9** | 59.6 |
| synth (D) | HL rand. | no col. | - | - | - | - | - | **43.6** | **68.6** | 54.1 | 79.8 | **60.0** |
| synth | learned | all | 53.2* | 77.2* | 53.1* | 78.9* | 58.0* | 33.0 | 58.2 | 51.8 | 78.0 | 56.7 |
| synth | HL learn. | no col. | **54.4** | **77.9** | **53.4** | **79.5** | **58.9** | 39.5 | 64.8 | 52.5 | 78.9 | 57.5 |
| synth (D) | HL learn. | no col. | - | - | - | - | - | **40.6** | **65.7** | **52.9** | **79.3** | **58.4** |

Table 3: Classification results obtained on top of the SimSiam pre-trained encoders. Bold indicates the best results for each group while underline the absolute best. **Left**: mean Top-1 accuracy for the BigBiGan encoders, averaged over the seven target datasets. **Right**: Top-1 and Top-5 accuracies for the StyleGan2 encoders on the two target datasets.

| Encoder | Mean Top-1 |
|---|---|
| Baseline real | **58.2** |
| Baseline synth | 47.2 |
| random | 47.0 |
| HL random | 59.6 |
| HL random (D) | **60.5** |
| learned | 46.2 |
| HL learned | 54.9 |
| HL learned (D) | **56.4** |

| Encoder | Target Dataset | | | |
|---|---|---|---|---|
| | Stanford Cars | | FGVC Aircraft | |
| | Top-1 | Top-5 | Top-1 | Top-5 |
| real | **33.4 ± 0.8** | **64.3 ± 0.4** | 20.7 ± 0.4 | 48.8 ± 1.1 |
| synth | 27.0 ± 0.2 | 54.6 ± 0.2 | **21.3 ± 0.7** | **50.5 ± 0.8** |
| random | 29.2 ± 0.4 | 58.1 ± 0.2 | 22.5 ± 0.6 | 51.7 ± 0.7 |
| HL rand. | **47.0 ± 0.3** | **76.1 ± 0.3** | **22.9 ± 0.8** | **53.5 ± 0.8** |
| learned | 28.6 ± 0.5 | 56.7 ± 0.2 | 22.0 ± 1.1 | 51.9 ± 0.5 |
| HL learn. | **35.2 ± 0.4** | **64.8 ± 0.5** | **23.0 ± 1.1** | **53.0 ± 0.9** |

In all experiments, HL perturbations outperform the baselines, proving the effectiveness of the proposed method. Where applied, discriminator filtering shows consistent improvements, suggesting that image quality plays a role but may not be a critical factor in achieving better representations. Interestingly, we observe that the HL *random* experiments often close the gap with the *learned* counterparts. Thus, applying hierarchical *random* perturbations can be a valid choice for training contrastive methods, without requiring any additional issue related to *walker* training. In comparison with real data, HL perturbations generally yield better or similar results. An exception occurs in the case of SimSiam encoders evaluated on ImageNet-1K, but we note that this gap narrows or disappears in other downstream tasks and datasets. We thus hypothesize that an encoder trained and evaluated on ImageNet-1K may exhibit a bias towards the limited training set, which does not affect the synthetic runs and disappears on other datasets.

**StyleGan2 and LSUN Cars.** In this setup, we train the encoders with a batch size of 512, due to StyleGan's larger number of parameters. All anchors are sampled from $\mathcal{N}^t(0., 1., 0.9)$, baseline positives from $\mathcal{N}^t(0., 0.15, 0.9)$ in *random* case, and use the 32K samples checkpoint in the *learned* case. For HL perturbations, we group the 16 chunks into 4 sets as discussed in Section 3.2. Specifically, we fix the second group and apply only small perturbations to the first one, giving more importance to the latter two. For *random*, we modify each group as $\{\mathcal{N}^t(0., 0.3, 1.), -, \mathcal{N}^t(0., 0.8, 1.), \mathcal{N}^t(0., 0.8, 1.)\}$, while the three trained *walkers* have seen 12K, 18K and 65K samples in the *learned* case.

Evaluations are performed on Stanford Cars Krause et al. (2013) and FGCV Aircraft 2013b Maji et al. (2013), running each experiment 5 times for 100 epochs, with a batch size of 256, SGD optimizer, learning rate of 30.0 and cosine decay. Average results are in Table 3 (Right), and show the superiority of HL perturbations over all baselines (including real data), confirming the findings observed before for BigBiGan, also in this fine-grained scenario. In particular, carefully designed hierarchical perturbations can lead to a great performance boost ($+13.6\%$ over the real baseline for HL *random* case on Stanford Cars). However, we note that *learned* perturbations struggle to follow

this trend, still surpassing the real case, probably due to the difficult optimization of Equation (3) in a fine-grained scenario.

## 5    RELATED WORK

**Positive views for contrastive learning.**    Due to the absence of annotated data, a key element of contrastive learning (Hadsell et al., 2006) lies in designing informative positive views Tian et al. (2020); Xiao et al. (2020). While methods like Bachman et al. (2019); Misra & Maaten (2020); Caron et al. (2020) used pretext tasks as matching global and local parts of an image, Chen et al. (2020b) introduced a set of effective manually designed data augmentations. Recently, efforts by Tamkin et al. (2020); Shi et al. (2022) improved upon these hand-crafted methods by learning good perturbations adversarially, while a growing number of approaches leveraged generators' latent spaces to sample positives (Yang et al., 2022b; Astolfi et al., 2023; Kim et al., 2023; Wu et al., 2023; Han et al., 2023) starting from real anchors. In contrast, the present work does not use real data, drawing both anchors and positives from the generator's latent space.

**Learning from generative models.**    Due to the growing performances of modern generative models like Rombach et al. (2022); Yu et al. (2022), several applications stand out. Examples include creating semantic-annotated datasets with minimal effort (Zhang et al., 2021; Melas-Kyriazi et al., 2021; Li et al., 2022a), or training supervised classifiers; both with synthetic data alone (Besnier et al., 2020; Sariyildiz et al., 2023; Lampis et al., 2023) or with a combination of real and synthetic data together (He et al., 2022; Bansal & Grover, 2023; Azizi et al., 2023). In representation learning, meaningful embeddings can be obtained through knowledge distillation (Yang et al., 2022a; Li et al., 2023b;a), or by sampling good anchors and positives in an unconditioned generator's latent space, as in Jahanian et al. (2021); Li et al. (2022b) or the present study. Recent works like Tian et al. (2023) consider instead a text-to-image setup, yielding promising results.

**Hierarchical-Latent models.**    The idea of introducing multiple hierarchical random vectors finds many implementations in the generative models' literature. Examples for Generative Adversarial Networks (GANs) (Goodfellow et al., 2014) include LapGan Denton et al. (2015), BigGans Brock et al. (2018); Donahue & Simonyan (2019) and the StyleGan family (Karras et al., 2019; 2020; 2021; Sauer et al., 2022) which employs a non-Gaussian latent space $\mathcal{W}$. Another notable work is Kang et al. (2023), which obtains state-of-the-art results in text-to-image synthesis. Variational Autoencoders (VAEs) Kingma & Welling (2014); Rezende et al. (2014) typically adopt HL structures to enhance the expressivity of the approximate distributions, as seen in NVAE Vahdat & Kautz (2020) and Ladder VAE Sønderby et al. (2016); Child (2020). Another notable work is Li et al. (2019), which extends the model of Zhao et al. (2017) to better capture the various disentanglement factors incorporated in the single hierarchies. Recently, promising advances have been made also in the field of Normalizing Flows Dinh et al. (2015); Rezende & Mohamed (2015), where Hu et al. (2022) draws inspiration from renormalization groups in physics to propose an HL architecture.

## 6    CONCLUSIONS

In this paper, we defined HL generative models, investigating how the multiple latent spaces can influence the image generation process. We also proposed a possible application of such models in the task of view generation for contrastive representation learning, showing better results with respect to state-of-the-art methods and real data baseline. Despite the existence of different implementations and numerous possible applications, to the best of our knowledge, the literature currently lacks a comprehensive theoretical framework for this family of generators, which we expect to become relevant in future research. Regarding the use of generators as a data source, we proposed continuous sampling as a way to increase the total training set size without requiring large storage capacities, reporting a comparable or shorter training time with respect to standard data loading techniques, at the price of a slightly higher $CO_2$ emissions. As generative models become an appealing alternative to standard datasets, we hope that future research will address these limitations, proposing not only faster but also less energy-demanding models.

## 7 Ethics Statement

The use of generative models as a data source may help in addressing several issues associated with real datasets. Specifically, it can prevent privacy and usage rights concerns related to genuine data (Kaissis et al., 2020; DuMont Schütte et al., 2021) or be used to censor sensitive attributes (Abbasi et al., 2021). On the other hand, it should be noted that these properties are not guaranteed, and generative models can be attacked Zhou et al. (2022) to leak information on the real data they were trained on Chen et al. (2020a); Nikolenko (2021). Moreover, since the biases of the original distribution can be inherited Asim et al. (2020), appropriate techniques to reduce these biases Tan et al. (2020); Teo et al. (2023) should be considered.

## 8 Reproducibility

The authors plan to release an open-source version of the training code, as well as pre-trained models upon paper acceptance. For all the experiments, details about relevant hyperparameters, like Gaussian distributions used for anchor and positive sampling or the specific checkpoints used for training the *walkers* are discussed in Section 4. Further implementation details, like the used pre-trained generators, code libraries, or data processing pipelines are reported in Appendix B.

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

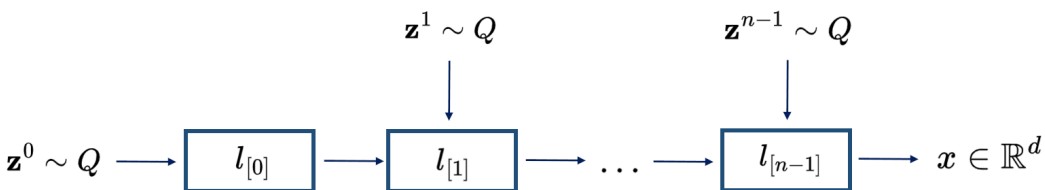

Figure 4: Schematic structure of a generic HL generative model with $n$ blocks $\{l_{[0]}, l_{[1]}, \ldots, l_{[n-1]}\}$ and input latent variables $\{\mathbf{z}^0, \mathbf{z}^1, \ldots, \mathbf{z}^{n-1}\}$, each sampled from a known distribution $Q$.

## A  HIERARCHICAL-LATENT GENERATOR STRUCTURE

In Figure 4 we present a schematic illustration of a HL generative model, using $n$ multiple latent variables $\{\mathbf{z}^0, \mathbf{z}^1, \ldots, \mathbf{z}^{n-1}\}$ to sample new data $x \in \mathbb{R}^d$. As explained is Section 3, each latent variable enters the network at a different progressive layer $\{l_{[0]}, l_{[1]}, \ldots, l_{[n-1]}\}$, thus providing a different level of contribution to the final sample.

## B  TRAINING DETAILS

All the code for this paper has been developed using `pytorch` Paszke et al. (2019) and `pytorch lightning` Falcon & The PyTorch Lightning team (2019). BigBiGan generator code and weights have been obtained at https://github.com/lukemelas/pytorch-pretrained-gans. For StyleGan2, the official `github` repositories are available, specifically for code at https://github.com/NVlabs/stylegan2-ada-pytorch and the weights at https://github.com/NVlabs/stylegan2. In the following, we report the training-specific details that have been used in the implementation.

**Data and preprocessing.**  All our experiments use FFCV Leclerc et al. (2023) for data storage and loading. For ImageNet-1K, images are stored at $256 \times 256$, and resized to $128 \times 128$ during loading, to match the output resolution of BigBiGan. Regarding LSUN Cars/StyleGan2, instructions on how to download and obtain the 893K training images can be found at https://github.com/NVlabs/stylegan2-ada-pytorch/tree/main. These are $512 \times 384$ images, which are stored at $512 \times 512$ with padding, to match the StyleGan2 outputs. During loading, images are first center cropped at $384 \times 384$, removing padding, and then resized at $128 \times 128$. The same process is repeated for the generated images.

For data augmentation/preprocessing we use the `kornia` library Riba et al. (2020). SimCLR transformations $T_\mathbf{x}$ are applied to the encoders as described in Section 4. In particular, we remove color jittering and grayscale from all our HL runs. During transfer classification learning, we apply random resize crop and random horizontal flip during training, and center crop during validation/testing. In all experiments, images are normalized with ImageNet mean and standard deviation values, and the final size (after cropping) is $112 \times 112$.

**Hardware resources and reproducibility.**  Most experiments have been run using 4 NVIDIA A100-SXM4-40GB GPUs, with an exception for the StyleGan trained encoders (which required 8) and some minor experiments like the *walkers* training, which used only 1. To ensure reproducibility, random seeds have always been fixed, and in particular during continuous sampling generations. This also allows for consistency throughout the synthetic runs, ensuring that each encoder sees the same images.

**Perturbations in $\mathcal{Z}$ and $\mathcal{W}$ space.**  In BigBiGan experiments, perturbations in the latent space are applied by summing the noise vector to the selected chunks, as described in Section 2. For StyleGan2, the final latent space is $\mathcal{W}$, and a mapping network is used to perform the operation $f(\mathbf{z}) = \mathbf{w}$. Here, $f$ is the mapping network, $\mathbf{w}$ the random latent vector in the $\mathcal{W}$ space, and $\mathbf{z}$ the initial random vector sampled from a truncated Gaussian distribution ($\mathcal{Z}$ space). The positive views in this case are obtained as $f(T_\mathbf{z}(\mathbf{z})) = \mathbf{w}$, where $T_\mathbf{z}$ is a *random* or *learned* perturbation that affects only the selected chunks.

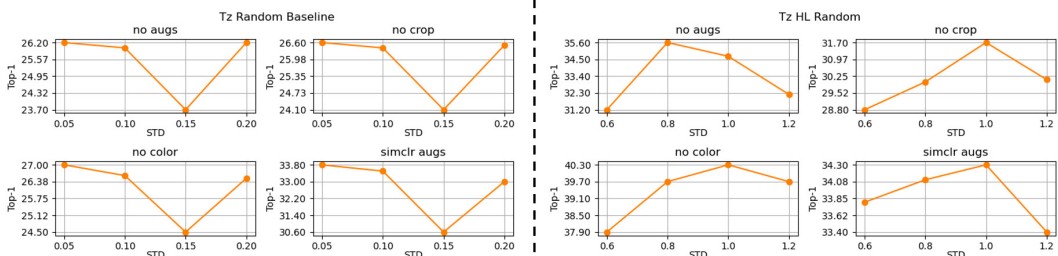

Figure 5: Ablation study for deciding $T_{\mathbf{x}}$ augmentations and standard deviation parameters for the *random* BigBiGan case. **Left**: for the *random* baseline, best results are obtained by sampling positives from $\mathcal{N}^t(0.0, 0.05, 2.0)$. Removing cropping or color augmentations causes a Top-1 accuracy drop of nearly 20% **Right**: for HL *random*, best results are obtained by sampling positives from $\mathcal{N}^t(0.0, 1.0, 2.0)$, with an exception when no $T_{\mathbf{x}}$ are applied, where a standard deviation of 0.8 achieves the best performances. Removing color augmentations allows the generation of more realistic views, boosting Top-1 accuracy by 15% with respect to the "SimCLR augs" case.

**Navigator training**  For the training of the non-linear *walkers*, we use the procedure introduced in Li et al. (2022b), where the specific checkpoint for each network is determined by monitoring the loss function, with some minor modifications. The code is available at https://github.com/LiYinqi/COP-Gen/tree/master. First, we initialize the weights using a Gaussian distribution $\mathcal{N}(0., 0.01)$ and the biases following a Uniform distribution $\mathcal{U}(-0.001, 0.001)$. This allows the training to start with an Identity mapping for $T_{\mathbf{z}}$. Second, we do not apply any pixel space transformation $T_{\mathbf{x}}$ during training, as expressed in Equation (3). This is to ensure a correct estimation for the $\delta$ values, which should be defined only with transformations in the latent space. Note that since $T_{\mathbf{z}}$ is initialized as the identity, the InfoNCE loss first reaches a minimum point, and then starts to increase. Therefore, we save the model weights when the loss reaches approximately the same value in all runs, identified as the point after which the generated views quickly change the image semantics.

All the *walkers* are trained with a batch size of 64, Adam Kingma & Ba (2015) optimizer with $\beta_1 = 0.5, \beta_2 = 0.999$ and a temperature $\tau = 0.1$ in the InfoNCE loss (Equation (1)). For single chunks or groups perturbations, the learning rate is $8 \times 10^{-5}$ for the *walker* and $5 \times 10^{-5}$ for the embedding function, while when training the *learned* baselines and the BigBiGan final *walker* we used a learning rate of $1 \times 10^{-5}$ for the *walker* and $3 \times 10^{-5}$ for the embedding function.

## C  $T_{\mathbf{x}}$ AUGMENTATIONS AND ABLATION STUDIES

To check which combinations of pixel space augmentations $T_{\mathbf{x}}$ to use, as well as to decide standard deviation hyperparameters and StyleGan2's baseline *walker* checkpoint, we perform several ablation studies with BigBiGan generator testing on ImageNet-100 and with StyleGan2 generator testing on StanfordCars (always using the SimSiam framework). The encoders use the same training procedures as detailed in Section 4, but with $\frac{1}{10}$ of the training steps. In general, we observe that all SimCLR augmentations are necessary to maintain good performances in the baselines, while color jittering and grayscale transformations can be removed in the HL counterparts. Detailed results are showed in Figures 5 to 7.

## D  TRANSFER LEARNING

We report in Table 4 the results for each transfer classification learning experiment performed on top of the SimSiam pre-trained encoders using BigBiGan / ImageNet-1K. Each experiment has been run with 5 different seeds, and the mean Top-1 accuracy is taken.

Regarding datasets, results are computed on the test set where available, otherwise on the validation set, maintaining the original splits. For DTD Cimpoi et al. (2014) the first proposed split has been used, while for Caltech101 Fei-Fei et al. (2004) we selected a random split of 30 train images per class, using the remaining for testing. All background images / distractors have been removed.

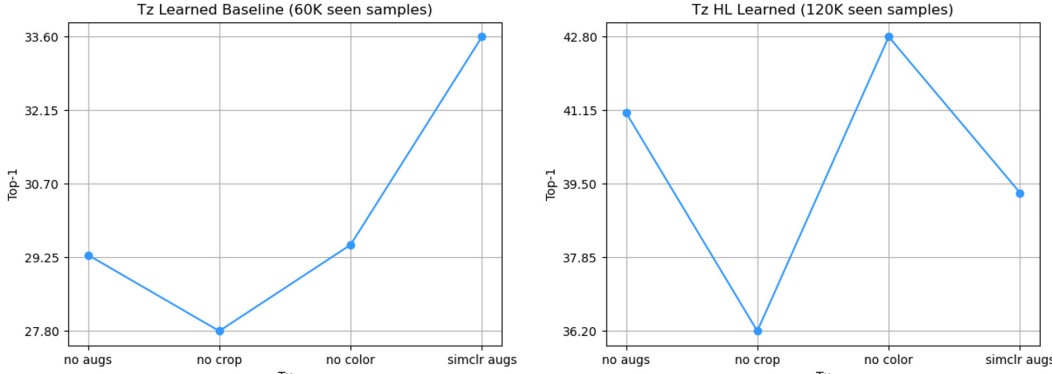

Figure 6: $T_{\mathbf{x}}$ augmentations on the selected *walker* checkpoints for baseline and HL BigBiGan experiments. **Left**: the checkpoint has seen 60K samples during training. All $T_{\mathbf{x}}$ augmentations are needed to obtain good results, with consistent drops in Top-1 accuracy if any augmentation is removed. **Right**: The selected checkpoint (120K training samples) achieves better results when color augmentations are removed. Interestingly, we note that applying all "SimCLR augs" is worse than applying no augmentations at all.

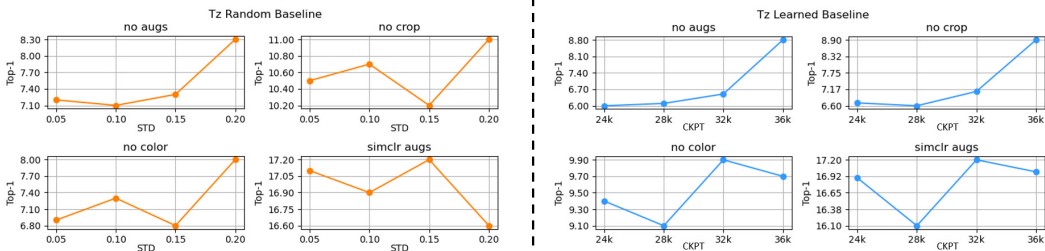

Figure 7: Ablation study for the *random* and *learned* baselines on the fine-grained StyleGan2 generator. **Left**: in the *random* case, best results are obtained by sampling positives from $\mathcal{N}^t(0.0, 0.15, 0.9)$ and applying all $T_{\mathbf{x}}$ augmentations. In general, we observe a great drop in performances when any of these transformations is missing. **Right**: Different checkpoints and $T_{\mathbf{x}}$ augmentations tested on the fine-grained StyleGan2's *learned* unitary baseline. The final selected checkpoint (32K seen samples) is the one achieving best Top-1 accuracy score in combination with all $T_{\mathbf{x}}$ transformations.

Table 4: Transfer classification Top-1 accuracy results for each pre-trained encoder using SimSiam framework and BigBiGan/ImageNet-1K as a data source. The first row presents a comparison with the supervised baseline. Experiments are obtained on 7 different target datasets.

| Encoder | Top-1 Accuracy on Target Dataset | | | | | | |
|---|---|---|---|---|---|---|---|
| | Birdsnap | Caltech101 | Cifar100 | DTD | Flowers102 | Food101 | Pets |
| Supervised Real | $93.6 \pm 0.5$ | $86.6 \pm 0.6$ | $48.1 \pm 0.7$ | $54.5 \pm 0.9$ | $79.1 \pm 0.7$ | $53.7 \pm 0.3$ | $84.9 \pm 0.5$ |
| Baseline real | $\mathbf{63.1 \pm 0.3}$ | $\mathbf{83.1 \pm 1.0}$ | $26.2 \pm 0.7$ | $\mathbf{56.4 \pm 0.3}$ | $\mathbf{59.8 \pm 2.6}$ | $\mathbf{51.5 \pm 0.2}$ | $\mathbf{67.6 \pm 0.4}$ |
| Baseline synth | $46.3 \pm 0.4$ | $67.9 \pm 1.6$ | $\mathbf{33.4 \pm 0.4}$ | $\overline{47.7 \pm 0.5}$ | $46.7 \pm 0.9$ | $41.6 \pm 0.2$ | $\overline{47.1 \pm 1.2}$ |
| random | $45.4 \pm 0.3$ | $68.9 \pm 0.9$ | $31.7 \pm 0.7$ | $47.9 \pm 0.6$ | $46.4 \pm 0.8$ | $42.3 \pm 0.3$ | $46.5 \pm 1.3$ |
| HL random | $64.3 \pm 0.6$ | $84.4 \pm 0.3$ | $\underline{41.1 \pm 0.8}$ | $54.9 \pm 1.0$ | $63.2 \pm 0.6$ | $50.1 \pm 0.3$ | $\mathbf{59.5 \pm 0.3}$ |
| HL random (D) | $\mathbf{65.2 \pm 0.2}$ | $\mathbf{85.1 \pm 0.3}$ | $40.4 \pm 0.2$ | $54.5 \pm 0.4$ | $\mathbf{66.8 \pm 0.6}$ | $\mathbf{52.0 \pm 0.3}$ | $59.5 \pm 1.0$ |
| learned | $42.0 \pm 0.3$ | $72.9 \pm 0.8$ | $31.7 \pm 0.8$ | $48.8 \pm 0.5$ | $45.2 \pm 0.7$ | $38.6 \pm 0.4$ | $44.0 \pm 0.8$ |
| HL learned | $57.7 \pm 0.5$ | $77.5 \pm 0.5$ | $\mathbf{36.7 \pm 0.4}$ | $53.1 \pm 0.4$ | $60.3 \pm 0.6$ | $47.6 \pm 0.5$ | $\mathbf{51.8 \pm 0.2}$ |
| HL learned (D) | $\mathbf{60.6 \pm 0.2}$ | $\mathbf{79.5 \pm 0.4}$ | $\mathbf{36.7 \pm 0.3}$ | $\mathbf{54.4 \pm 0.8}$ | $\mathbf{62.6 \pm 0.8}$ | $\mathbf{50.2 \pm 0.5}$ | $51.0 \pm 0.6$ |

For completeness, in the first row of Table 4 we also report Top-1 accuracy obtained from the supervised baseline. In this case, the ResNet-50 encoder has been trained on ImageNet-1K for 100 epochs, achieving a Top-1 accuracy of $60.1\%$ on the test set. Transfer learning results are obtained using the same linear classification protocol denoted in Section 4.2.

## E    DETAILS FOR DISCRIMINATOR FILTERING

Some of our experiments aim to explore how the quality of generated images affects overall performance. While prior work (Li et al., 2022b) proposed a comparison with a more powerful generator (Casanova et al., 2021), we instead maintain the same model (BigBiGan) and enhance data quality through discriminator filtering. Specifically, we first train a small discriminator network for one epoch to distinguish real (ImageNet-1K) from fake (BigBiGan) images, assigning a score ranging from 0 (fake) to 1 (real). During experiments conducted with SimSiam, each batch is sampled twice, and the discriminator assigns scores to each image. Only the top half (those with higher scores) are retained and used as the final input batch. Each discriminator input is pre-processed with a small Gaussian Blur (kernel size 5, $\sigma = 1.5$ using `kornia`) and normalized with ImageNet mean and standard deviation values.

## F    EXAMPLES OF GENERATED VIEWS

Figures 8 and 9 report some examples of generated views with our method and the corresponding *random* and *learned* baselines. All views are obtained with the same hyperparameters reported in Section 4 and used for training the contrastive encoders.

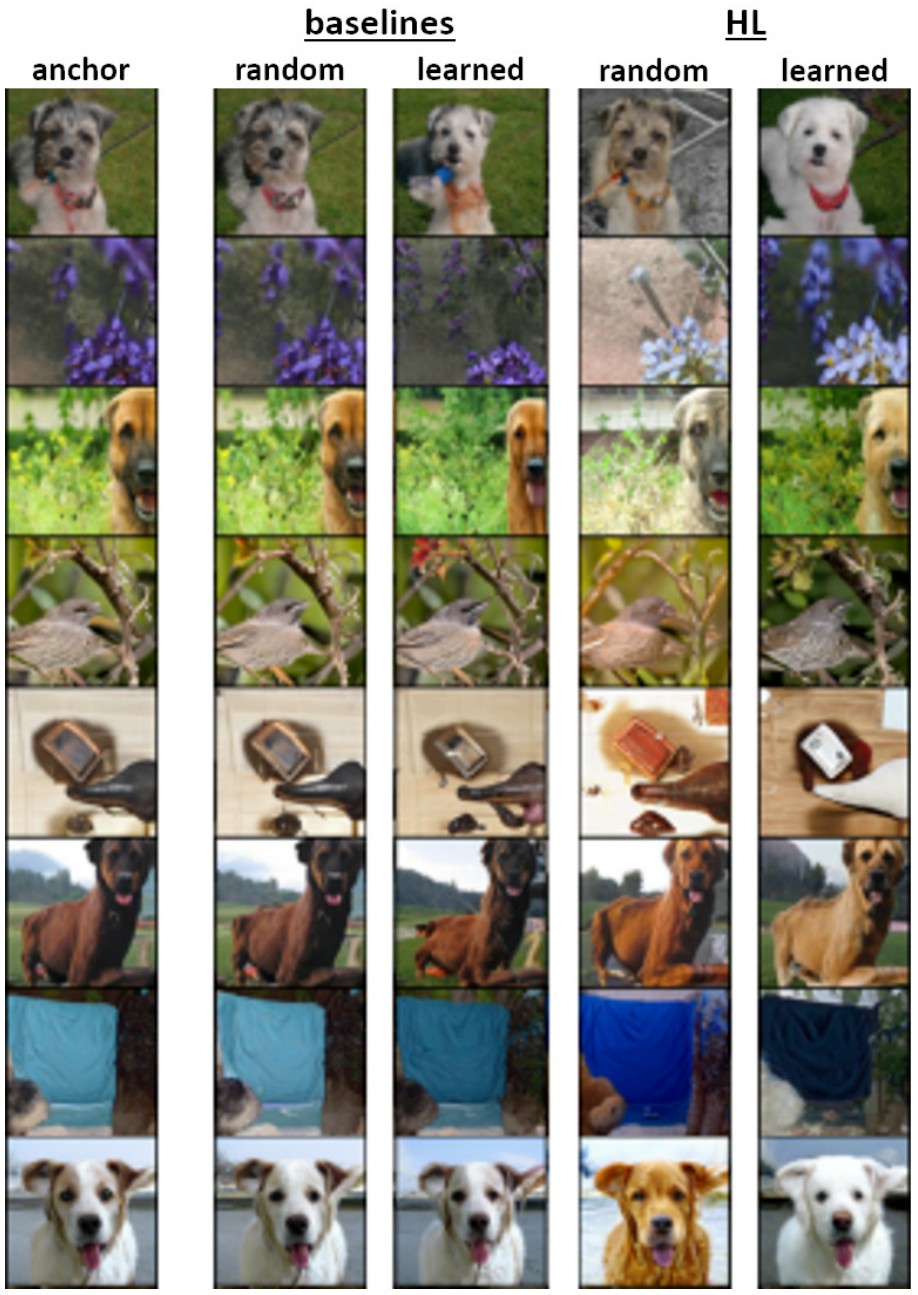

Figure 8: Example of generated views from BigBiGan anchors. From left to right: anchor, *random* and *learned* baseline views, *random* and *learned* HL views.

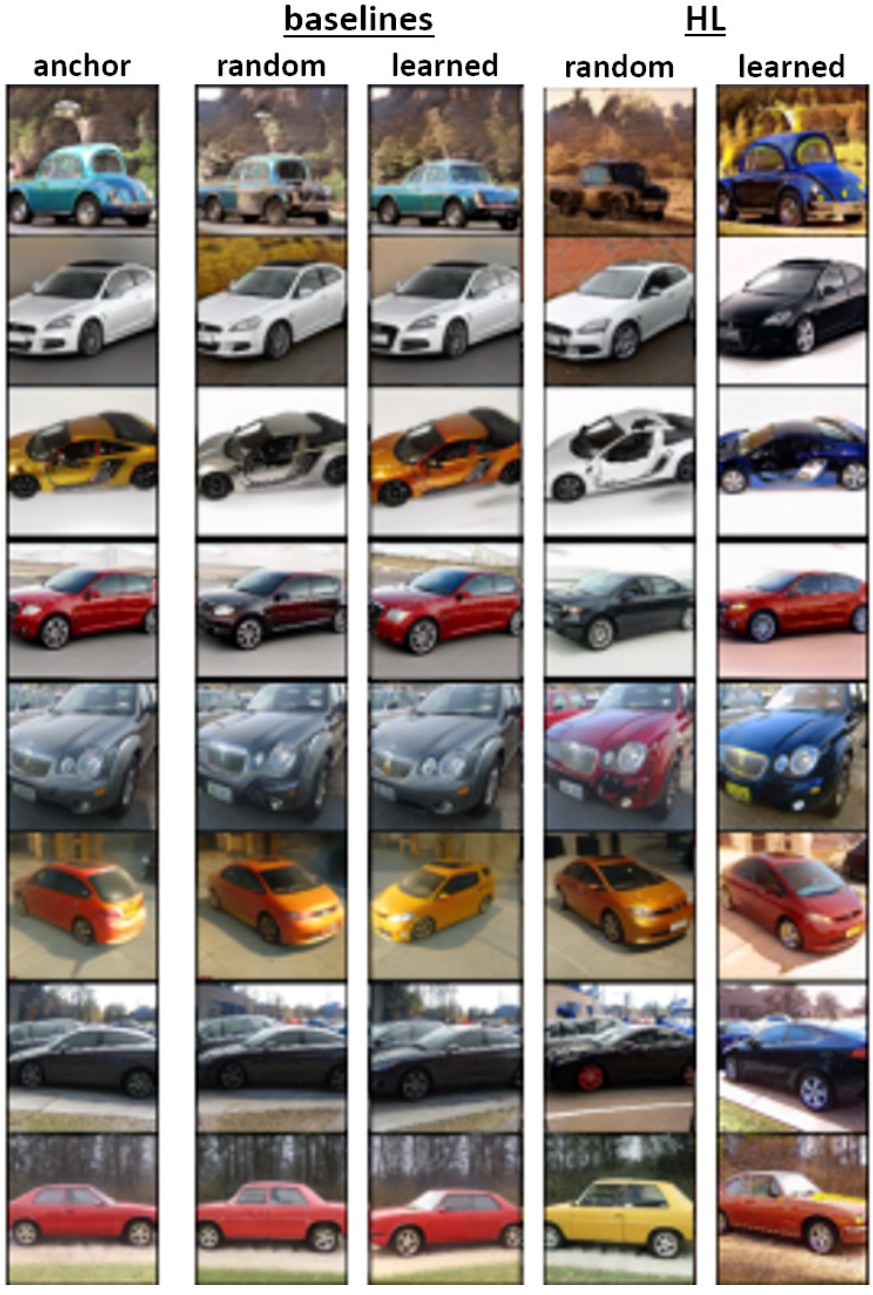

Figure 9: Example of generated views from StyleGan2 anchors. From left to right: anchor, *random* and *learned* baseline views, *random* and *learned* HL views.

