# OpenReview forum: "Hierarchical-Latent Generative Models are Robust View Generators for Contrastive Representation Learning"
_ICLR.cc/2024/Conference — ICLR 2024 Conference Withdrawn Submission_

### Official Review · Reviewer_5q8s · 2023-10-15

**Soundness:** 2 fair
**Presentation:** 3 good
**Contribution:** 2 fair
**Rating:** 5
**Confidence:** 3

**Summary:**

In this paper, the authors introduce and explore a category of generative models termed Hierarchical-Latent (HL) models. This work demonstrates how the distinctive characteristics of these models, which operate using multiple latent spaces in a hierarchical fashion, can be harnessed to generate robust positive views for contrastive representation learning. By employing HL models as data sources for self-supervised learning and devising specific perturbation strategies for different latent vectors, the authors achieve significant improvements in representation learning compared to previous methods and even surpass the performance of training on real data. Additionally, this study proposes a continuous sampling approach for generating additional training data in real time, revealing its competitive or faster performance compared to traditional data loading techniques. Overall, the paper formalizes the HL model category, highlights its effectiveness in self-supervised learning, and introduces a practical method for augmenting training data.

**Strengths:**

(1) This paper introduces the new concept of Hierarchical-Latent (HL) models and their application in self-supervised learning. The formalization of HL models as a distinct category within generative models, along with their utilization for data generation, presents a fresh perspective.

(2) The paper meticulously conducts experiments, compares the proposed approach with existing methods, and includes comprehensive ablation studies to validate its efficacy. Furthermore, the introduction of the continuous sampling technique to augment training data is practical.

(3) The paper is clear in presentation. This work effectively describes the concepts, with a logical flow from defining HL models to their application in self-supervised learning. The use of illustrative figures enhances comprehension.

(4) This paper holds significance in the fields of self-supervised learning and generative models. The proposal of continuous sampling as a means to augment training data online addresses a pertinent issue, potentially reducing the gap between synthetic and real data in classifier training.

**Weaknesses:**

(1) The paper lacks a deeper theoretical analysis of why and how the Hierarchical-Latent models would significantly enhance the paper. Providing theoretical insights into the relationships between hierarchical latent, perturbations, and representation learning could strengthen the paper's contributions.

(2) This work lacks a comprehensive comparison with strong baseline methods. A more extensive set of baseline models and techniques should be considered to provide a more thorough evaluation of the proposed approach.

(3) Visualization of the hierarchy of latent spaces in HL models or diagrams illustrating this hierarchical structure would enhance understanding.

(4) A more in-depth discussion of the trade-offs and limitations of using generative models for self-supervised learning compared to real data would be valuable.

**Questions:**

(1) Could the authors provide more theoretical insights into the hierarchical nature of HL models' latent spaces and their implications for representation learning?

(2) Could the authors expand the baseline comparisons by including more state-of-the-art self-supervised learning methods and possibly discussing how the proposed approach compares to conventional supervised learning using real data?

(3) Would it be possible to include visualization to illustrate the hierarchical structure of latent spaces in HL models?

(4) Have the authors examined or experimented with the adaptability of HL models and the suggested approach in fields beyond images, such as text or audio data?

---

> ### Author Response · Authors · 2023-11-14
> **Theoretical insights into the relationships between hierarchical latent, perturbations, and representation learning.**
>
> As per the relation between latent perturbations and the generation of views for representation learning, the used procedure (Eq. 3 of our paper) points at maximizing the distance between the two latent vectors $d(\mathbf{z}, T_\mathbf{z}(\mathbf{z}))$, while maximizing the mutual information between the generated views $g(\mathbf{z}), g(T_\mathbf{z}(\mathbf{z}))$. As demonstrated by Oord et al. (2018), maximizing a lower bound on mutual information is equivalent to minimizing InfoNCE Loss, which is the term optimized in Equation 3. This procedure has been previously introduced and theoretically motivated in Li et al. (2022b).
>
> In our work, we extended such findings and procedures to generative models presenting multiple layers of latent variables, which we name Hierarchical Latent. In detail, we show a relation between the latent variables' hierarchical level and the amount of perturbation that can be applied to them, while still maintaining semantic consistency.
>
> In the updated version of the paper, we included some insights in the paragraph above Equation 3, to better explain why InfoNCE loss can be used to find optimal latent transformations in the context of representation learning.
>
> _References_
>
> Li et al. (2022 b): Yinqi Li, Hong Chang, Bingpeng Ma, Shiguang Shan, and Xilin Chen. Optimal positive generation via latent transformation for contrastive learning. Advances in Neural Information Processing
> Systems, 35:18327–18342, 2022b.
>
> Oord et al. (2018) Aaron van den Oord, Yazhe Li, and Oriol Vinyals. Representation learning with contrastive predictive
> coding. arXiv preprint arXiv:1807.03748, 2018

---

> ### Author Response · Authors · 2023-11-14
> **Requests for more experiments comparing with different Self-Supervised methods and Supervised Training**
>
> We would like to point out that the main scope of the paper is to show the benefits that a smart use of Hierarchical Latents can provide in the context of __view generation__ for representation learning. Under this perspective, the baselines to consider are methods that use a generative model for view generation. To the best of our knowledge, the only two methods that actually perform such task are Li et al. (2022b) and Jahanian et al. (2021), which are extensively compared in the experimental sections. It is also valuable to note that we already extended the benchmark proposed by these papers, which solely utilized SimCLR as the self-supervised framework. For this purpose, all the baselines were re-implemented and further evaluated using the SimSiam framework, as reported in the right part of Table 2.
>
> As for the comparison with supervised methods using real data, we included new results as an additional row in Table 4 of the Appendix. In detail, results are computed for the same ResNet50 backbone, trained on Imagenet-1k in a supervised manner and evaluated on all the transfer classification datasets presented in the paper. As expected, supervised training performs better than self-supervised. This is true for both the source dataset (Imagenet-$1$K) and most of the transfer target datasets. We thank the Reviewer for the suggestion, which broadens the experimental evaluation.
>
> _References_
>
> Li et al. (2022 b): Yinqi Li, Hong Chang, Bingpeng Ma, Shiguang Shan, and Xilin Chen. Optimal positive generation via latent transformation for contrastive learning. Advances in Neural Information Processing
> Systems, 35:18327–18342, 2022b.
>
> Jahanian et al. (2021): Ali Jahanian, Xavier Puig, Yonglong Tian, and Phillip Isola. Generative models as a data source
> for multiview representation learning. In International Conference on Learning Representations,
> 2021

---

> ### Author Response · Authors · 2023-11-14
> **Other Questions (points 3 and 4)**
>
> 3. __About visualization of the hierarchical structure of latent spaces.__ We added a section in the Supplementary material (appendix A), where a schematic structure of a generic HL generative model is given. We hope that this can help in better understanding the HL models architecture. We also point the Reviewer to Brock et al (2018) and Karras et al (2020), where complete pictures of the architectures used can be found.
>
> 4. __About the adaptability of HL models on other modalities.__ We thank the reviewer for the precious suggestion, which we did not consider in our paper and poses an interesting basis for possible future research directions. In our work, we considered image data since HL generative models are well established in such domain. In fact, images present a global-to-local structure that well fits the HL framework. Interestingly, similar structures can be found also for audio data. For example, Hono et al. (2020) proposed a HL model for speech synthesis. On the other hand, we did not find similar architectures for text data, where generative models do not usually consider a hierarchical structure with multiple input latent vectors.
>
> _References_
>
> Brock et al. (2018): Andrew Brock, Jeff Donahue, and Karen Simonyan. Large scale gan training for high fidelity natural
> image synthesis. In International Conference on Learning Representations, 2018
>
> Karras et al. (2020): Tero Karras, Samuli Laine, Miika Aittala, Janne Hellsten, Jaakko Lehtinen, and Timo Aila. Ana-
> lyzing and improving the image quality of stylegan. Proceedings of the IEEE/CVF conference on
> computer vision and pattern recognition, pp. 8107–8116, 2020.
>
> Hono et al. (2020): Hono, Y., Tsuboi, K., Sawada, K., Hashimoto, K., Oura, K., Nankaku, Y., Tokuda, K. (2020). Hierarchical multi-grained generative model for expressive speech synthesis. arXiv preprint arXiv:2009.08474.

---

### Official Review · Reviewer_DrZA · 2023-10-28

**Soundness:** 3 good
**Presentation:** 3 good
**Contribution:** 3 good
**Rating:** 6
**Confidence:** 2

**Summary:**

This paper discusses the use of hierarchical-latent generative models as robust view generators for contrastive representation learning. The authors propose a framework that utilizes the properties of hierarchical-latent models to create robust views for contrastive learning, outperforming previous methods and even surpassing approaches trained with real data. The model is novel and the performance is shown to be better than the current baselines.

**Strengths:**

(1) The problem that the paper aims to solve is significant and the method propose is feasible.
(2) The paper forms the problem in a theoretical way, which is technically sound.
(3) The performance of the proposed method is superior based on the experimental section.

**Weaknesses:**

(1) In proposition 2.2, the paper says g(.) is a mapping to x but in the equation it says g(.) maps to y instead.
(2) The paragraph above assumption 3.1, the paper says the last layer model fine-grained details of the data, but I think this can also be done but the last layer of any neural networks, such as MLP.
(3) I would also suggest the paper to add complexity analysis to their method as this might be a concern if too many latent variables have to be generated.

**Questions:**

Please refer to my comments under "Weakness".

---

> ### Author Response · Authors · 2023-11-14
>
> 1. __Notation in Proposition 2.2__: The purpose of the $\rightarrow$ operator in the Proposition was to denote that the two generated images $g(\mathbf{z}_i, \pmb{\theta})$ and $g(\mathbf{z}', \pmb{\theta})$ are mapped into the same semantic label $\mathbf{y}$.
> We acknowledge that it is ambiguous, and thus we changed it in the newly uploaded version of the paper:
>
> \\[
> \delta_i = \max_{\mathbf{z'}} \text{dist} (\mathbf{z}_i, \mathbf{z'}) \quad \text{ such that } \quad F\_{\mathcal{T}}(g(\mathbf{z}_i, \pmb{\theta})) = F\_{\mathcal{T}}(g(\mathbf{z'}, \pmb{\theta})) = \mathbf{y}
> \\]
>
> where $\mathbf{z'} \in \mathbb{R}^n$ is a generic vector in the generator's latent space and $F_{\mathcal{T}}$ is an oracle classification function for task $\mathcal{T}$, which assigns the true semantic label $\mathbf{y}$ to any image.
>
> 2. __Paragraph above Assumption 3.1__: The reviewer is right. In fact, in the paragraph and in Definition 3.1, we are not assuming any particular internal structure for the generator network $g$, which could possibly be a simple MLP for simpler tasks. What we want to stress here is that latent variables $\{ \mathbf{z}^0, \mathbf{z}^1, \dots, \mathbf{z}^{n-1} \}$ operate at progressive layers as inputs of the generator. This makes their contributions different, since later layers model more fine-grained details of the data.
>
> 3. __Complexity__: our method does not introduce any additional complexity, which only depends on the choice of the generator architecture. In our work, we utilize two well-known Generative Adversarial Networks (BigBiGan and StyleGan2), which, as any GAN, present the advantage of a constant inference time. We take advantage of this fact by also proposing _continuous sampling_, as outlined in the last part of Section 4.1.

---

### Official Review · Reviewer_o5fU · 2023-10-30

**Soundness:** 1 poor
**Presentation:** 3 good
**Contribution:** 3 good
**Rating:** 3
**Confidence:** 3

**Summary:**

This paper focuses on using image generation models for contrastive self-supervised learning. Specifically, it employs a hierarchical structure of multiple latent vectors in the generative models to change the global to local information of generated images. New images are generated based on progressive perturbations of the latent vectors through hierarchies. The key message is that as the level in the hierarchy increases (from global latent variables to local ones), the perturbation distances in the latent space to change the semantics of generated images increase (i.e., at more local levels of latent vectors, changing the semantics of generated images requires more aggressive perturbation.) The authors then use the hierarchical regime to generate images for self-supervised learning and propose to continuously sample generated images to increase the total training size. The authors show SOTA results on training SimCLR and SimSiam using BigBiGan and StyleGAN2 to generate images.

Despite good efforts, the current shape of the paper lacks important technical details to make it sound and clear enough to be accepted at ICLR.

**Strengths:**

Originality: this work starts from an interesting perspective: using hierarchical levels of latent codes to generate images for representation learning and study the effect of perturbation in terms of the maximum distance allowed in the latent space to avoid semantic change. This is new to the reviewer’s knowledge.

Clarity: overall, the paper is clearly written, with an easy-to-follow presentation. The theoretical results are clearly stated, generated samples are nicely illustrated, and empirical results are concisely presented.

Significance: using generative models as data sources is an interesting and important problem, and this paper provides potentially useful techniques and insights for future researchers.

**Weaknesses:**

Quality: there are severe gaps in the theoretical part despite the adequate empirical results and analyses. The details are in the questions section.

**Questions:**

1. **Proposition 2.2**: it is very unclear how the authors deduced such an important statement, which itself also has ambiguity. The authors claim Proposition 2.2 is a reformulation from Li et al. (2022b), but there are no proofs or theoretical discussions that support this (upon the reviewer’s inspection, there is no directly equivalent statement in Li et al. (2022b)). In Li et al. (2022b), semantic consistency is defined as a bounded difference between mutual information terms, where the mutual information is between the generated images and the label. However, none of these is clearly stated in this submission, nor are there any discussions or linkages. In terms of the ambiguity, what does the right arrow mean? Is it convergence in probability, in distribution, or something else? Why does such convergence (or the mathematical property the authors intended to show) suffice the expressing the semantic consistency?

2. **Table 1**: it is quite debatable if InfoNCE loss is a good metric to measure semantic shift/consistency of images, which is also a central component this work’s conclusions rely on (Table 1 directly supports Assumption 3.1 empirically, which is the key message of the paper and does not have other rigorous theoretical justification). The authors did not justify at all why InfoNCE loss can align well with the intention of Proposition 2.2 regarding Semantic Consistency. In Proposition 2.2, semantic consistency is defined in terms of the consistency of true labels. However, the authors did not show how InfoNCE can *preserve* or approximate such consistency property when labels are not present. The absence of task information/labels does not automatically justify InfoNCE as the valid metric.

Other questions or comments:
1. It is not very clear to the reviewer why, in Table 1 left, different chunks use a different number of training samples for the Monte Carlo simulations.
2. Whether “robust” is the best word for this submission is debatable when it essentially means task-agnostic.

---

> ### Author Response · Authors · 2023-11-14
> **Notation and doubts about Proposition 2.2**
>
> we acknowledge that the meaning of the $\rightarrow$ symbol is ambiguous. It was meant to denote that the two generated images $g(\mathbf{z}_i, \pmb{\theta})$ and $g(\mathbf{z}', \pmb{\theta})$ are mapped to the same semantic label $\mathbf{y}$. For this reason, the symbol has been removed and replaced with a new formulation:
>
> __Proposition 2.2__: (Semantic Consistency in the Latent Space - (reformulation))
>
> Let $g(\mathbf{z}, \pmb{\theta}) = \mathbf{x}$ be a generative model with parameters $\pmb{\theta}$ that maps from latents $\mathbf{z} \in \mathbb{R}^n$ to images $\mathbf{x} \in \mathbb{R}^d$. Consider a downstream task $\mathcal{T}$ with label $\mathbf{y} \in \mathcal{Y}$, and some distance metric *dist* defined in the generator's latent space.
> Then, for any random latent vector $\mathbf{z}_i$, the term $\delta_i$ indicates the maximum distance in the latent space of $g$ to ensure semantic consistency for task $\mathcal{T}$:
>
> \\[ \delta_i = \max_{\mathbf{z'}} \text{dist} (\mathbf{z}_i, \mathbf{z'}) \quad \text{such that} \quad  F\_{\mathcal{T}}(g(\mathbf{z}_i, \pmb{\theta})) =  F\_{\mathcal{T}}(g(\mathbf{z'}, \pmb{\theta}))  =  \mathbf{y} \\]
>
> where $\mathbf{z'} \in \mathbb{R}^n$ is a generic vector in the generator's latent space and $F_{\mathcal{T}}$ is an oracle classification function for task $\mathcal{T}$, which assigns the true semantic label $\mathbf{y}$ to any image.
>  __********__
>
> Regarding the connection with Proposition 3.1 of Li et al. (2022b), it states:
>
> "Given a well-trained unconditional generative model $g$, $\mathbf{z}$ and $\mathbf{z}'$ are two latent vectors, __if the distance__ $d(\mathbf{z}, \mathbf{z}') \le \delta$, __then__ $g(\mathbf{z})$ __and__ $g(\mathbf{z}')$ __will have the similar semantic label__ $\mathbf{y}$.
>
> In the form of mutual information, we have:
>
> \\[
>     |I(g(\mathbf{z}); \mathbf{y}) - I(g(\mathbf{z}' ); \mathbf{y})| \le \epsilon
> \\]
>
> where $\epsilon$ stands for tolerable semantic difference, and $\delta$ __is the maximum shifted distance to maintain semantic consistency__."
>
> In our proposed reformulation (Proposition 2.2), 1) we omitted the mutual information part since not in the scope of our work: we are mostly interested in defining what the term $\delta$ represents, that is, the semantic consistency in the latent space of a generator $g(\mathbf{z}, \pmb{\theta})$; 2) we formalize the concept of "__having the similar semantic label__ $\mathbf{y}$" by means of the oracle classifier $F_{\mathcal{T}}$.
>
> In conclusion, __both propositions state__ that for each latent vector $\mathbf{z}_i$, it exists a $\delta_i$ ensuring that an image $g(\mathbf{z}', \pmb{\theta})$ shares the same semantic label as $g(\mathbf{z}_i, \pmb{\theta})$, as long as $d(\mathbf{z}_i, \mathbf{z}') \le \delta_i$.
>
> _References_
>
> Li et al. (2022 b): Yinqi Li, Hong Chang, Bingpeng Ma, Shiguang Shan, and Xilin Chen. Optimal positive generation via latent transformation for contrastive learning. Advances in Neural Information Processing
> Systems, 35:18327–18342, 2022b.

---

> ### Author Response · Authors · 2023-11-14
> **Table 1 and InfoNCE loss**
>
> As expressed in Section 3.2, the results of Table 1 are obtained utilizing the procedure reported in Eq. 3, which uses InfoNCE as a mutual information estimator. This procedure has been introduced and used by Li et al. (2022b) (the same formula is presented in Equation 6 of their paper). In the same work, a theorem and formal proof are given, supporting the validity of the procedure.
>
> In short, the equation (their Eq. 6) consists of a min-max game between the minimization of InfoNCE Loss and the maximization of $d(\mathbf{z}, T_\mathbf{z}(\mathbf{z}))$. In this context, the minimization of InfoNCE Loss corresponds to a maximization of the Mutual Information between views, making it a valid choice for our purposes. This property of InfoNCE loss __is demonstrated in Oord et al. (2018)__
>
> We thank the Reviewer for raising this point. In the new version of the paper, we modified the paragraph above Equation 3, in order to stress the relation between mutual information and InfoNCE loss.
>
> _References_
>
> Li et al. (2022 b): Yinqi Li, Hong Chang, Bingpeng Ma, Shiguang Shan, and Xilin Chen. Optimal positive generation via latent transformation for contrastive learning. Advances in Neural Information Processing
> Systems, 35:18327–18342, 2022b.
>
> Oord et al. (2018) Aaron van den Oord, Yazhe Li, and Oriol Vinyals. Representation learning with contrastive predictive
> coding. arXiv preprint arXiv:1807.03748, 2018

---

> > ### Comment · Reviewer_o5fU · 2023-11-17
> > **Response to the authors**
> >
> > The reviewer thanks the authors for their substantial efforts to improve the draft and clarify the questions.
> >
> > InfoNCE is a lower bound of mutual information between views, but why it is a good measurement in Table 1 for semantic consistency / semantic shift with respect to labels is not theoretically justified. If the reviewer understands correctly, from Li et al. (2022 b), semantic consistency in latent space holds *only if* $d\left(\mathbf{z}, T_{\mathbf{z}}(\mathbf{z})\right) \leq \delta$. Then, we can create optimal views and learn representation by maximizing InfoNCE bound (or equivalently minimizing the loss). This means semantic consistency with respect to labels is deduced from $d\left(\mathbf{z}, T_{\mathbf{z}}(\mathbf{z})\right) \leq \delta$, not simply InfoNCE. In an oversimplified fashion, it seems that the logic is $d\left(\mathbf{z}, T_{\mathbf{z}}(\mathbf{z})\right) \leq \delta \rightarrow \text{semantic consistency} \rightarrow \text{InfoNCE}$.
> >
> > However, in Table 1, InfoNCE is *directly* used to measure the semantic consistency/shift *without* any condition from $d\left(\mathbf{z}, T_{\mathbf{z}}(\mathbf{z})\right)$. It is even the other way around: InfoNCE becomes a measurement to indicate whether different $d\left(\mathbf{z}, T_{\mathbf{z}}(\mathbf{z})\right)$ preserves semantic consistency. In this case, an oversimplified logic is $\text{InfoNCE} \rightarrow \text{semantic consistency} \rightarrow d\left(\mathbf{z}, T_{\mathbf{z}}(\mathbf{z})\right) \leq \delta$. This is quite different from the first logic shown above.
> >
> > Why InfoNCE can measure semantic consistency w.r.t. labels is not clear. An example would be two images: one depicting a small seagull in the sky and the other depicting a small sailboat in the sea. There can be substantial mutual information between the two structurally and pixel-wise similar images (a largely blue background with a small white object at the center), but the labels are different. The reviewer would appreciate any further clarification from the authors.

---

> > > ### Author Response · Authors · 2023-11-17
> > > **On semantic consistency in Li et al. (2022 b)**
> > >
> > > Dear reviewer,
> > > thank for your comment, which helped us clarifying the InfoNCE issue.
> > >
> > > Concerning you first point, in order to define semantic consistency Li et al. 2022b explicitly say that
> > >  > if the distance $dist(\mathbf{z}, \mathbf{z}') \le \delta$, then $g(\mathbf{z}; \pmb{\theta})$ and $g(\mathbf{z}'; \pmb{\theta})$ will have the __similar semantic label y__
> > >
> > > Please note that this is not an _only if_ statement.
> > >
> > > Before moving to the InfoNCE loss, Li et al. define
> > > \begin{equation}
> > >             |I \bigl(g(\mathbf{z}; \pmb{\theta}); \mathbf{y}) \bigr) -  I \bigl(g(\mathbf{z}'; \pmb{\theta}); \mathbf{y}) \bigr) | \le \epsilon
> > >         \end{equation}
> > > as a tolerable semantic distance.
> > >
> > > In other words, $\delta$ denotes the maximum distance between two latent vectors for which the generated views maintain a similar level of information towards the semantic label $\mathbf{y}$, according to this equation.
> > >
> > > However, in the context of Self-supervised Learning, the information about the semantic label $\mathbf{y}$ is __unknown__, since it depends on the specific downstream task. For this reason, Li et al. 2022b implicitly relax the constraint above and minimize the InfoNCE loss between two views (more in detail the InfoNCE is calculated between two feature representations of the views, as in equation 3 of our paper).
> > > \begin{equation}
> > >           \mathcal{L}_{NCE} ( f(g(\mathbf{z}; \pmb{\theta}) ), f(g(\mathbf{z}'; \pmb{\theta}) ) )
> > >         \end{equation}
> > > In this respect InfoNCE can be regarded as a good proxy for semantic consistency. Yet, having no labels available for assessing semantic consistency, this is the best proxy we can have, although not perfect, as in the seagull/sailboat example. Being just a proxy, there is no hard logical implication that can formally be established but only a loose connection.
> > >
> > > To conclude, the InfoNCE values reported in Table 1 are those corresponding to different $\delta$, This is just to show that in order to obtain the same semantic shift (amount of mutual information between views), one needs to move more in the latent space of higher chunks.
> > >
> > > We hope that this makes thing clearer!
> > > Thank you again for your time and for the fruitful discussion.

---

> ### Author Response · Authors · 2023-11-14
> **Further Questions or Comments**
>
> 1. Table 1 shows that different chunks require a different number of training samples for the Monte Carlo simulations to achieve a similar InfoNCE loss. The intuition is that walkers $T_\mathbf{z}$ responsible for fine grained details (i.e. acting on high level chunks) must be trained a lot more to obtain a level of perturbation consistent with previous walkers (i.e. low level chunks). Note that each training starts with the same hyperparameters (optimizer, learning rate) and is initialized as the identity mapping function (at training step 1 we have $T_\mathbf{z}(\mathbf{z}) = \mathbf{z}$).
>
> 2. The term task-agnostic can indicate almost any self-supervised learning method, since in general the downstream task is always unknown (as also in Li et al. (2022b)). However, we show in the experimental results that our method is robust across a range of downstream tasks, thus the term "robust" is used.
>
> _References_
>
> Li et al. (2022 b): Yinqi Li, Hong Chang, Bingpeng Ma, Shiguang Shan, and Xilin Chen. Optimal positive generation via latent transformation for contrastive learning. Advances in Neural Information Processing
> Systems, 35:18327–18342, 2022b.

---

### Author Response · Authors · 2023-11-14
**General Reply to All Reviewers**

We thank all Reviewers for the time spent reading our work. We have uploaded a revised version of our paper, where we addressed the raised remarks, which helped us in upgrading substantially our work. In short, modifications include some changes to Proposition 2.2 and the text of Section 2 (as suggested by Reviewers o5fU and DrZA). In the Appendix, some experiments have been added to Table 4, as well as a new Section containing a schematic illustration of a Hierarchical-Latent model. These changes address the suggestions proposed by Reviewer 5q8s. For further details, please see the specific answers we provided to each Reviewer.